# Incorporation of MyDispense, a Virtual Pharmacy Simulation, into Extemporaneous Formulation Laboratories

**DOI:** 10.3390/healthcare10081489

**Published:** 2022-08-08

**Authors:** Joseph A. Nicolazzo, Sara Chuang, Vivienne Mak

**Affiliations:** Faculty of Pharmacy and Pharmaceutical Sciences, Monash University, Parkville, VIC 3052, Australia

**Keywords:** extemporaneous formulation, MyDispense, virtual simulation, medicines dispensing

## Abstract

A core competency of Australian Pharmacy graduates is to prepare and compound extemporaneous formulations. Students in our pharmacy course would traditionally formulate extemporaneous products in laboratory classes while simultaneously preparing a handwritten label, with students divorcing this laboratory activity from the entire dispensing process. As a way to incorporate the dispensing process into the preparation of extemporaneous products without adding excessive time to the laboratory, we integrated MyDispense, a virtual pharmacy simulation, in pre-laboratory activities. This meant that students could complete all the dispensing activities for prescribed extemporaneous formulations prior to attending the laboratory. Prescriptions for solutions, suspensions, creams and ointments were developed in MyDispense, including essential components for dispensing an extemporaneous formulation (e.g., formulation name, dosing instructions). These prescriptions were provided to students at least 1 week prior to their laboratory classes, whereas for the laboratory assessments, the prescription was provided at the commencement of the extemporaneous exam. Due to the implementation of dispensing via MyDispense, we found that students demonstrated pre-laboratory engagement as all students presented their printed labels upon entering the laboratory. We also observed an increase in interaction between students and laboratory facilitators, mainly focused on the principles of formulation integrated around patient outcomes. Virtual simulations such as MyDispense can therefore provide a guided realistic learning experience, whilst overcoming time pressures associated with laboratory timetabling. This approach also encourages students to engage in the dispensing process prior to extemporaneous laboratories providing more opportunity to discuss higher-level formulation principles and patient-centred outcomes during laboratory classes.

## 1. Introduction

Dispensing of medicines is a core component of the role of a pharmacist, a competency that is essential to ensure that medicines are provided to patients in a safe and accurate manner. Some of these medicines need to be extemporaneously prepared, such as creams, ointments, solutions and suspensions. It is a requirement of the Australian Pharmacy Council that all students graduating from a registered Pharmacy program in Australia or New Zealand have demonstrated competency in preparing and supplying extemporaneously compounded medicines safely and accurately [1]. The dispensing of an extemporaneously prepared formulation adds a layer of complexity to the general dispensing process, as this requires a pharmacist to draw on their fundamental knowledge of factors affecting drug and excipient solubility and stability, while still garnering medication history and providing patient counselling. Students have a demonstrated understanding of the relevance and importance of extemporaneous formulation in a clinical setting [2,3], and approaches aimed at enhancing activities associated with extemporaneous formulation, such as the inclusion of additional pre-laboratory workshops, have been shown to improve student outcomes in assessments [2]. It is therefore essential for Pharmacy students to obtain the necessary training in both the dispensing and preparation of extemporaneous products simultaneously. With limited laboratory time and space in many Pharmacy courses, intelligent ways to maximize time in the allocated laboratory, while still ensuring the dispensing process is front and centre of students’ minds, is essential for curricular design and student outcomes. Such examples that have been previously employed to assist in providing additional extemporaneous formulation exposure outside of the laboratory include the use of videos [4,5], while in one report, a mobile app was used to provide not only instructional videos, but also notes, chemical information and graphical representations of the laboratory [6]. However, it should be noted that most of the focus of these approaches have been on laboratory skills, which, while beneficial, did not focus on relevant patient aspects important in the dispensing process.

Given that students must demonstrate competency in the preparation and supply of extemporaneously prepared products, we considered approaches that would maximize time in the allocated laboratories for “hands on” activity, without jeopardizing the students’ opportunity to experience all elements of the dispensing process. This philosophy was supported by feedback from key stakeholders including community pharmacists, intern training program providers and experienced teaching associates, which highlighted that students appeared to divorce the preparation of extemporaneous formulations from the overall dispensing process (from prescription receipt to patient counselling), often not relating the prepared formulation to the prescribed patient. Furthermore, stakeholder feedback suggested that students did not appear confident to prepare, record and dispense new formulations, which is aligned with that reported in previous studies [2].

## 2. MyDispense: Incorporation of Dispensing into the Preparation of Extemporaneously Prepared Products

The development of a new vertically integrated Pharmacy degree at Monash University, which focused on enhancing various student skills and self-directed learning [7], was an opportune time to develop approaches to maximize “hands on” time in the laboratory for the preparation of extemporaneous formulations without segregation of these activities from the entire dispensing process. This was made possible through the virtual pharmacy simulation, MyDispense. MyDispense is a virtual pharmacy simulation where students obtain the necessary practice to dispense medicines in a safe environment [8] and which has garnered extensive uptake across Pharmacy educators worldwide [9,10,11,12,13,14]. The virtual simulation has an inbuilt number of diverse patients with various medical conditions and histories, and students are able to practice dispensing a number of exercises that allow them to make decisions to dispense or not dispense based on the patient’s medical history and other medicine-related information. In addition to providing a safe environment for dispensing skill development, MyDispense has been successfully implemented in self-care therapeutics courses [15], in applying controlled substance laws in the dispensing process [16,17], in preparing students for community pharmacy practice experiences [18] and to increase student confidence in over-the-counter recommendations [19]. While the use of this simulation has enhanced the student experience with non-extemporaneous dispensing, we were the first to design activities in MyDispense that could be employed for extemporaneous pharmacy activities. We developed a number of virtual scenarios in MyDispense for students to engage in prior to the extemporaneous laboratories, allowing more time for actual “hands on” activity within the laboratories. Each week, students were required to dispense a prescription for an extemporaneous formulation based on commonly compounded formulations listed in the Australian Pharmaceutical Formulary. Over a period of 4 weeks, students had the opportunity to dispense (and subsequently formulate) either a solution, suspension, cream or ointment, with each formulation dispensed (and prepared) in the same week that the theory and application of that formulation was covered in pre-lecture and lecture material. In detail, (i) students accessed the relevant case study on the MyDispense Dashboard and reviewed instructions on the introduction screen; (ii) they reviewed the prescription presented by the patient for safety and clinical appropriateness; (iii) if safe and appropriate, the role of each excipient within the prescribed formulation was checked based on prescribed resources; (iv) the prescription was dispensed for the right patient, and a virtual label was generated with the appropriate directions; and (v) students printed a physical copy of the label to be presented at the laboratory. As MyDispense had never been used to generate physical labels, a printable labelling platform was built, allowing students to print their label at home, which they would then bring to the relevant laboratory. With this novel approach, students were therefore exposed to the entire process of dispensing from prescription receipt (prior to the laboratory) to formulation preparation (within the laboratory). New prescription recording documentation was also developed for students to pre-populate prior to the laboratory class, so that students could (i) identify the role of every excipient based on their formulation science instruction that week, (ii) plan the extemporaneous method they would implement (based on descriptions provided in the Australian Pharmaceutical Formulary), (iii) pre-select necessary ancillary labels, and (iv) prepare patient counselling points in advance. This was a mandatory element to be granted access into the laboratory, so that students were as prepared as possible for the preparation of the extemporaneous formulation when they entered the laboratory.

## 3. Outcomes from Incorporating MyDispense into the Preparation of Extemporaneous Formulations

We noted complete student engagement, with 100% of students always presenting their printed labels upon entering the laboratory. In addition to providing more time for “hands on” extemporaneous activity, an unexpected consequence of dispensing prior to entry into the laboratory was that students appeared to interact with each other and their facilitators with higher level questions around formulation components. This suggested that dispensing their prescription in a virtual environment and completing part of their batch and prescription record in advance may have more effectively prepared them for the laboratory. What was extremely pleasing to see was that students were more commonly discussing patient-specific issues when preparing extemporaneous products, suggesting that this new approach was indeed placing the patient at the centre of student learning. These informal observations of student discussion around higher level formulation principles and patient specific factors are expected to have improved student competencies around formulation preparation and patient counselling, albeit this was not formally evaluated. The incorporation of MyDispense into the extemporaneous laboratories also provided students with an opportunity to practice their skills in written communication (through generating labels with clear and concise directions) and inquiry (through identifying the role of each excipient): skills which are an integral component of our new integrated Masters Pharmacy degree [7].

## 4. Employing MyDispense for Extemporaneous Laboratory Examinations

While students were able to dispense and prepare their label prior to normal laboratory classes, this was not possible for the laboratory examination, which occurred 2 weeks after the final laboratory class. For this, the examination feature in MyDispense was employed, where students were presented with the prescription and patient-relevant information at the commencement of an invigilated laboratory examination. Students were provided up to 30 min to prepare and submit the label, which was saved within the simulation software. Each label was saved with a personalized student identifier, allowing the examiner to batch print the labels and then provide the correct label to the correct student in the laboratory based on this personalized identifier. Immediately following submission of their label in MyDispense, students were able to commence the preparation of their assigned extemporaneous product. They were then required to present their final product with the printed label (and any ancillary labels) affixed to the container alongside their completed batch and prescription record. In addition to the accuracy of their label and identifying relevant medicine and patient-specific issues, students were assessed on the final product formulated, the accuracy of the details provided in their batch and prescription record, formulation technique and counselling points.

## 5. Future Prospects

There are many opportunities for which MyDispense can be augmented to enhance learning around dispensing extemporaneous formulations. The ability for students to practice compounding in a hybrid format allows them to focus on the hands-on extemporaneous skills in the laboratories, especially when there are limited space resources and time sensitivity. MyDispense can be further developed by building upon these elements of extemporaneous dispensing through inclusion of other elements already developed in MyDispense, i.e., through combining the dispensing of extemporaneous products with experiential education and preparation for clinical placements, teaching of law instruction and self-care practice [12]. Such future activities may therefore further enhance the translatability of MyDispense into the real-life scenarios that are encountered in pharmacy.

It would have been ideal to compare the outcomes of student cohorts who had employed MyDispense within the extemporaneous formulation examination with those who did not have access to MyDispense. However, others have demonstrated that the use of MyDispense has increased student perceptions and knowledge [19], albeit in the area of over-the-counter medicine dispensing, and others have also demonstrated that the inclusion of additional material (e.g., videos) together with extemporaneous laboratories has resulted in improved student confidence and competence [4]. Further studies to compare student performance in extemporaneous formulation examinations with and without the inclusion of MyDispense is critical to evaluate the effectiveness of this approach on student learning and performance. Furthermore, a multischool evaluation of the effectiveness of MyDispense in enhancing student learning in extemporaneous formulation is critical to robustly demonstrate the benefit of MyDispense in this educational activity.

## 6. Summary

This is the first report of implementing MyDispense into the dispensing and production of extemporaneous formulations. The successful outcomes of including MyDispense into extemporaneous formulation activities and the findings of this work may be of benefit to other Pharmacy programs worldwide given that this approach can provide more “hands on” laboratory time with better links to patient-centred care. The importance of placing the dispensing process at the forefront of students’ minds prior to preparation of extemporaneous products has been informally shown to enhance student understanding of formulation principles, as they relate to patients, which can be of benefit to other Pharmacy programs worldwide.

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
