# Peer review of "Incorporation of MyDispense, a Virtual Pharmacy Simulation, into Extemporaneous Formulation Laboratories"

_healthcare, 2022, doi:10.3390/healthcare10081489_

Round 1
Reviewer 1 Report
The authors could improve the paper by adding the below points:
1. Could be improved by adding the process flow of MyDispense.
2. Could add on the part related to - which types of skills improvement in students?
3. What will be the future prospects of MyDispense?
Reviewer 2 Report
Dear authors, I was very interested reading your commentary. Every new or modified step in pharmacist education it is of great importance for further improvement of pharmacy profession. It would be great if simulation you have used can be implemented worldwide.Since this article is commentary, the main question was ragarding the implementation of the new approach into the education of pharmacy students.
1.Was this step valuable considering the maximization of the education process regarding the competences of preparing the extemporaneously products and patient counselling upon its completion?
2.Also, was it a good guidline for pharmacy schools worldwide?
3.At last, but the least important was it a good example of implementing a new techonologies into pharmacy education?
4.Originality is daubtful in its origin, but still for them it is new, at my department we have something similar but not with software. Cinsidering some practice around world it is theirs approach presented to other. It is not new idea, but original as the whole process?
5. I think authors provide an experience, but as well positive practice that can be further implemented. Considering other articles, we have some reports from US. As I said it was not a new idea, it is their approach with possibilty to be acceptedmby others. I understand the commentary should be understand like that.
6. little bit more about MyDispence would be appriciated. Can you describe more detailed myDispence? Should you extend the conclusion, with little more focused on education process and would you be a good example for the pharmacy schools worldwide.
Reviewer 3 Report
Thank you for the opportunity to review this commentary. The authors identify use of a virtual pharmacy simulation software as a potential tool for increasing engagement in compounding laboratories, and support that aim with brief anecdotal evidence. The commentary is well written and organized, though I believe it is lacking in relevant literature support and more detailed scholarly evidence of the tool’s effectiveness, which may limit its wider appeal to the readership. My comments below represent minor revisions that if completed would improve the publishability of the commentary.
Introduction:
-
Line 37-40: The authors identify the complexity of extemporaneous compounding, but do not support this with any literature. Including literature highlighting student perceptions of compounding, student performance in compounding, etc would add more weight to your commentary.
-
Line 46: What literature exists for current creative/innovative use of laboratory time? There may not be specific articles relative to compounding but there are several that highlight unique strategies (many you highlight later on that I think would be worth including a sentence or two to indicate lack of virtual technology in compounding)
-
Line 52-55: You identify students' perceptions around compounding from your own stakeholder experience. How is this supported by the literature?
Outcomes:
-
As a whole, this section is lacking. Is there quantitative evidence you can include to support your qualitative claims? Comparing performance / activity scores from previous years would provide stronger support for the effectiveness of the virtual activity
-
Line 108-110: If no data is available for your cohort of students, can you support your results with data from literature referencing virtual learning combined with hands on? Even beyond pharmacy education - other disciplines often utilize virtual learning to provide needed educational outcomes. The outcomes you identify here need to be supported by data and as written, there is little to indicate applicability across other cohorts/Schools
Summary:
-
Would like to see more discussion around next steps or further areas of focus. Are there other utilities for the MyDispense software that the authors could see it being expanded into? Are there other online technologies that could be utilized?
